# Dynamic structure of *E. coli* cytoplasm: supramolecular complexes and cell aging impact spatial distribution and mobility of proteins
Dmitrii Linnik, Ivan Maslov, Christiaan Michiel Punter & Bert Poolman ✉

Protein diffusion is a critical factor governing the functioning and organization of a cell's cytoplasm. In this study, we investigate the influence of (poly)ribosome distribution, cell aging, protein aggregation, and biomolecular condensate formation on protein mobility within the *E. coli* cytoplasm. We employ nanoscale single-molecule displacement mapping (SMdM) to determine the spatial distribution of the proteins and to meticulously track their diffusion. We show that the distribution of polysomes does not impact the lateral diffusion coefficients of proteins. However, the degradation of mRNA induced by rifampicin treatment leads to an increase in protein mobility within the cytoplasm. Additionally, we establish a significant correlation between cell aging, the asymmetric localization of protein aggregates and reduced diffusion coefficients at the cell poles. Notably, we observe variations in the hindrance of diffusion at the poles and the central nucleoid region for small and large proteins, and we reveal differences between the old and new pole of the cell. Collectively, our research highlights cellular processes and mechanisms responsible for spatially organizing the bacterial cytoplasm into domains with different structural features and apparent viscosity.

Protein motion is a key feature of all forms of life and is essential for the functioning of cells. Prokaryotes generally lack membrane-bound organelles and active intracellular transport. Most of the biochemical reactions take place in a continuous cytoplasm, lacking membrane sub-compartments, and depend on Brownian motion for the biomolecules to interact[1,2]. Protein lateral diffusion depends on the hydrodynamic radius of the molecules and the viscosity of the medium. We have previously shown that the mobility of a diverse set of native and foreign proteins in *Escherichia coli* scales with the mass of the protein complexes with a power law dependence[1]; similar observations have been made by others[3–5]. The lateral diffusion in the cytoplasm deviates from the Einstein-Stokes relationship and we concluded that proteins perceive an apparent viscosity that varies with their molecular mass[1]. We also observed for the entire set of tested proteins a significant decrease of the lateral diffusion coefficient ($D_L$) in the pole areas of the cells, and each cell having a "fast" and "slow" pole[1,2]. In this work we investigate the molecular mechanisms underlying the differences in the mobility of proteins present in different regions of the cell.

Molecules diffusing inside the cell can perceive distinct local microenvironments of biomolecular condensates, or membrane-less organelles formed via liquid-liquid phase separation (LLPS, Fig. 1a)[6]. The LLPS can occur as a part of a normal cell growth or can be induced by stress conditions. In healthy cells, biomolecular condensates are formed e.g by pole-organizing protein PopZ[7], cytoskeletal protein FtsZ, and nucleoid-exclusion protein SlmA[8]. In starved cells, biomolecular condensation drives the assembly of PolyP granules[9]. Biomolecular condensates can be composed of proteins and DNA (RNAP condensates on promoter sequences[10] and ParB clusters on *parS* sites[11]), proteins and RNA (BR-bodes controlling mRNA degradation[12]), or can be formed by natively-disordered regions in membrane proteins in close proximity of the cell membrane (FHA domains of the ATP-binding cassette transporter Rv1747 from *M. tuberculosis*[13]). If such condensates would have preferred intracellular localization, e.g. cell poles as observed for PopZ[7], this could lead to an apparent decrease of the lateral diffusion coefficient.

The nucleoid of the bacterial cell is a membrane-less organelle that allows passage of most macromolecules but excludes translating ribosomes (polysomes, Fig. 1b). The polysomes are abundant at the poles and periphery of the cell[14,15]. Aggregates of misfolded or damaged proteins and

Department of Biochemistry, University of Groningen, Groningen Nijenborgh 4, 9747 AG, the Netherlands. ✉e-mail: b.poolman@rug.nl

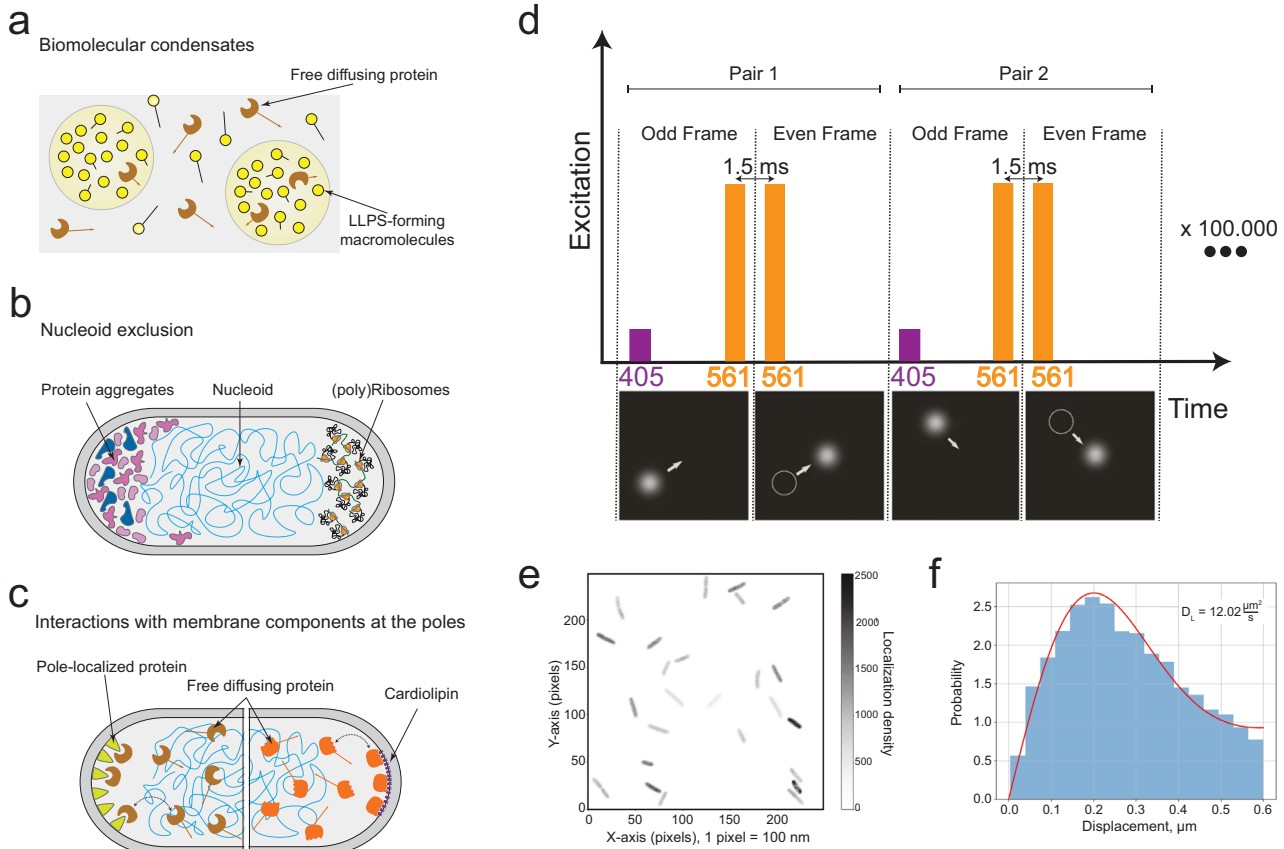

**Fig. 1 | Formation of subcellular domains with different protein mobility/viscosity and description of SMdM procedure. a** Liquid-liquid phase separation (LLPS) can lead to the formation of biomolecular condensates and create regions in the cytoplasm with different protein mobility (indicated by the length of the arrows). **b** Exclusion of macromolecular complexes (such as protein aggregates and poly-ribosomes) from the nucleoid lead to areas in the cytoplasm with different crowding and macromolecular viscosity. **c** Transient membrane binding of molecules (shown by two-sided arrow) via specific proteins or lipids (e.g. cardiolipin) residing at the poles of the cell. **d** SMdM procedure using stroboscopic illumination. Short low-intensity 405 nm (purple) laser pulses are spaced at the beginning of odd frames, which convert mEos3.2 from a green to red fluorescent state. 561 nm (orange) laser pulses excite red fluorescent mEos3.2 at the end of odd frames and beginning of even frames with pulse-to-pulse time separation of 1.5 ms. This pattern of frame pairs is repeated for 100.000 frames totally. **e** 250 by 250 pixels field of view to depict all single-molecule localizations as two-dimensional histogram. **f** Distribution of measured displacements presented as a histogram and fitting of the data with adjusted probability density function of a 2-dimensional random-walk diffusion model.

plasmids are also excluded from the nucleoid[16,17]. Their abundance is increasing under stress conditions, such as heat shock[18,19], nutrient starvation, or dysregulation of protein synthesis[20], but they have also been found in non-stressed cells[21]. Asymmetric distribution of protein aggregates is associated with aging of bacterial cells[21,22], an inherited characteristic of binary cell division[23].

Binary division of bacteria is geometrically symmetrical, but it is not functionally symmetrical: the daughter cells are morphologically and genetically identical, but they differ in cytoplasmic and membrane content[24]. As an example, IbpA, a small heat shock protein, associated with utilization of protein aggregates, localizes predominantly at the old pole of the cell[21]. *E. coli* daughter-cells originating from the "old" pole of the mother-cell exhibit a diminished growth rate (decreased metabolic efficiency), decreased offspring biomass production, and also increased chance of death, as compared to daughter cell originating from the "new" pole[25]. The inherited and accumulated protein aggregates are, likely, one of the major causes of cell aging in binary-dividing cells[2].

Protein mobility can also be decreased by intermolecular interactions within the cytoplasm or with components of the cell membrane (Fig. 1c). Peptidoglycan synthesis takes place in the middle part of the cell, from where the "old" cell wall is continuously pushed toward the poles[26,27]. This pole-oriented motion of the peptidoglycan guides the directed motion of membrane proteins[14]. Moreover, the altered curvature of the membrane at the cell poles may cause the accumulation of specific proteins and lipids at

these sites[28,29]. Hence, the content of the pole membrane differs from the rest of the cell, and the presence of specific factors in the membranes at the poles can act as a driving force for the formation of cytosolic pole-localized protein pools. The localization of the integral membrane proteins ProP[30] and MscS[31] is thought to be driven by interaction with cardiolipin, which is most abundant at the poles[29]. Also proteins of the Min system interact preferably with cardiolipin[32,33], which facilitates the formation of the FtsZ ring in the middle of the cell.

We determine protein mobility in *E. coli* using single-molecule displacement mapping (SMdM)[1,34]. This recently developed technique uses photoactivated localization microscopy (PALM) for the localization of individual fluorescently-labeled proteins on consecutive frames of microscopy recording, while they diffuse inside a cell (Fig. 1d-f). Unlike conventional single-particle tracking PALM (spt-PALM), SMdM employs stroboscopic illumination to reduce the motion blur typical of spt-PALM data; the method ensures high spatial resolution of the diffusing particles. In contrast to for instance, fluorescence correlation spectroscopy (FCS) and fluorescence recovery after photobleaching (FRAP), SMdM allows simultaneous acquisition of data in various regions of a cell or several adjacent cells, which allows the detection of a large number of molecules with a high precision and high throughput. Using SMdM, we evaluate how polysome distribution, cell aging, the presence of biomolecular condensates (specifically, PopZ condensates), and protein aggregation affect the diffusion of proteins within a cell.

## Results

### Ribosome distribution is not affecting protein diffusion at the cell poles

(Poly)ribosomes assemble at the periphery of the cell and are excluded from the nucleoid region, whereas ribosomal subunits can diffuse through the entire cytoplasm[15]. Hence, the polyribosomes could potentially contribute to slower diffusion of proteins in the pole regions of the cell, which we observed in our previous study[1]. To test this hypothesis, we measured the protein diffusion in *E. coli* cells treated with erythromycin and rifampicin.

Erythromycin (Ery) binds to the 50 S ribosome subunit and prevents the elongation of the peptide chain. Rifampicin (Rif) inhibits bacterial RNA polymerase and affects protein synthesis by blocking the production of mRNA thus preventing polyribosome formation[15,35]. Lower abundance of mRNA in the cytoplasm of rifampicin-treated cells also leads to increased protein mobility[36]. To confirm redistribution of ribosomes by antibiotic treatment we fused *rpsB* (one of the ribosome small subunit proteins gene) to the gene for mRuby3 fluorescent protein and expressed the construct from the chromosome of *E. coli* BW25113[37]. The growth rate of BW25113-*rpsB::mRuby3* was indistinguishable from the wildtype strain (Fig. S1). Widefield fluorescence microscopy of the RpsB-mRuby3 fusion showed a decrease of ribosome abundance at the cell poles in cells treated with 50 or 250 ng/mL of erythromycin and almost no pole or peripheral clustering of ribosomes in cells treated with 500 ng/mL of rifampicin (Fig. 2a, b). These data suggest that a fraction of the ribosomes dissociates in the presence of erythromycin, whereas the (poly)ribosomes disassemble completely, due to loss of mRNA, in the presence of rifampicin.

Using SMdM, we measured the lateral diffusion coefficient ($D_L$) of photoconvertible fluorescent protein mEos3.2 in the cytoplasm of cells treated with erythromycin and rifampicin as well as non-treated cells. In untreated and antibiotics-treated cells, the diffusion of mEos3.2 is significantly slower at the cell poles compared to the mid-cell region (Fig. 2c). Erythromycin treatment (250 ng/mL; Fig. 2d) caused a slight but significant increase in $D_L$ in the middle of the cell. The pole/middle ratios of $D_L$ remain the same (Fig. 2c), while the changes in ribosome distribution are significant (Fig. 2b). Rifampicin treatment leads to a 32% increase in $D_L$ in the middle of the cell ($p < 0.0001$), and a 10% increase ($p < 0.0001$) in the pole regions, compared to solvent-treated control cells (Fig. 2d). The relative difference of the diffusion coefficient between cell poles and mid-cell regions is even more pronounced in rifampicin-treated than in control cells ($p = 0.0381$, Fig. 2c). Taken together these data indicate that the presence of polysomes at the cell poles is not a major cause for the lower mobility, because the difference in pole/middle $D_L$ ratio is also found in cells where the ribosome subunits are evenly distributed: The pole/middle $D_L$ ratio and the difference between $D_L$ in the middle and at the poles do not correlate with the distribution of ribosomes (Fig. S2). We attribute the overall increase in protein mobility to a decreased effective viscosity of the cytoplasm as a result of mRNA depletion upon rifampicin treatment.

### Cell aging is affecting diffusion at the pole regions of E. coli

We then investigated the effect of cell aging on the diffusion in different regions of the cell. After cell division there is an asymmetry in the distribution of cytoplasmic components as a large fraction of the molecules outside the nucleoid is displaced towards the poles of the mother cell, which becomes the old pole of daughter cells[23,24]. Hence, the majority of the supramolecular complexes and aggregates of misfolded or damaged proteins will end up in the old pole[22].

We measured lateral diffusion of mEos3.2 in the *E. coli* cytosol in the terminal stages of cell division and immediately after the division, as in this case it is possible to keep track of the old and new pole (Fig. 3a). The mobility of mEos3.2 was significantly lower at the old pole compared to the new pole of the cell (Fig. 3b). Analysis of the two-by-two contingency table with "fast/slow" pole and "old/new" pole as parameters also show a significant correlation (Fig. 3b, bottom panel). We also observed an unequal distribution of mEos3.2 localizations in the cell: The central part of the cell (60% of the volume) had 78% of the displacements and 73% of localization data, and the

new and old poles (each 20% of the volume) had 14% and 8% of the displacements and 9% and 16% of the localizations, respectively (Fig. S3).

Pole-organizing protein Z (PopZ) is a native protein of *Caulobacter vibrioides* that plays a role in cell division and interacts with ParB-*parS* complexes[7]. The full-length PopZ expressed in *E. coli* also clusters at the old pole of the cell where it forms a phase-separated condensate with the ParB protein and *parS*-centromere like sequence[7]. To distinguish the old and new cell poles via widefield fluorescence microscopy prior to our SMdM measurments, we constructed an *E. coli* strain expressing mEos3.2 and PopZ-eGFP (Fig. 3c). We observed an overall decrease in mobility (at old pole, new pole and cell middle) of mEos3.2 when PopZ was overexpressed, which may reflect an overall increase in macromolecular crowding (Fig. 3f, Fig. S4). The mobility of mEos3.2 was significantly lower at the old (PopZ-positive) pole compared to the new (PopZ-negative) pole of the cell (Fig. 3d, e). The two-by-two contingency also shows significant correlation between pole age and $D_L$ decrease (Fig. 3d, bottom panel). The trends in the pole/middle $D_L$ ratios are the same when the PopZ localization or cell division is used to discriminate the old and new pole (Fig. 3e). The measured $D_L$ and pole/middle $D_L$ ratios are summarized in Fig. 3f. We propose that the decrease of $D_L$ at the *E. coli* poles and differences between old and new pole of the cell are caused by protein aggregates, which accumulate more at the old than new pole. Representative probability density function fitting profiles of the displacements in different regions of the cells are shown in Fig. S5.

### Mass-dependent behavior of SMdM probe

To test whether the decrease of the protein mobility at the cell poles depends on the size of the probe, we measured $D_L$ of three different mEos3.2 fusions: AceB-mEos3.2 (malate synthase A, $M_W = 85.9$ kDa), Icd-mEos3.2 (isocitrate dehydrogenase, $M_W = 142.8$ kDa), and IlvC-mEos3.2 (ketol-acid reductoisomerase, $M_W = 318.9$ kDa). To exclude possible effects of (poly)ribosomes we treated cells with rifampicin. Representative probability density fittings of displacements for these proteins in the middle part of the cell are shown in supplementary Fig. S6. We observe a strong correlation ($r^2 = 0.95$ for a linear fit) between the molecular weight $M_W$ of the SMdM probe and the pole/middle $D_L$ ratios, which increases from 0.69 to 0.78 in the $M_W$ range from 25.7 to 318.9 kDa (Fig. 4b–d).

### Induced protein aggregation slows down the lateral diffusion

To investigate the effect of protein aggregates on the protein mobility at the cell poles, we induced protein aggregation by heat-shock stress[38]. *E. coli* cells were grown at 30 °C for 4 h until $D_{600} \sim 0.15$ and then transferred to 42 °C for one hour before SMdM. Brightfield images of heat-shocked cells show light-dense aggregates at either one (~75% of measured cells) or both (~25% of measured cells) cell poles (Fig. 5a). In the heat-shocked cells with aggregates at both poles, the pole/middle $D_L$ ratios decrease at both the "slow" (old) pole and "fast" (new) pole as compared to non-shocked cells. In the heat-shocked cells a single aggregate is always localized at the "slow" (old) pole. The pole/middle $D_L$ ratio in these cells decreases only for the "slow" (old) pole (Fig. 5b). This means that there is a difference in protein aggregate formation in the old and new pole of the cell, for which the high-$M_W$ meshed structures may act as nucleus for aggregation. Displacement and diffusion maps also showed fewer displacements and decreased mobility in pole areas with visible aggregates (Fig. 5c). In fact, in cells with aggregates at the poles the number of detected displacements in the regions close to the cell edge was often too low to construct diffusion maps as seen for the right pole of the cell presented in the right panel of Fig. 5c.

### Diffusion in cells with inhibited division machinery is not homogeneous and shows alternating patterns of diffusivity

The location of protein aggregates in *E. coli* changes when cell division is disrupted. Cells with compromised division (with deletion of *minCD* genes) have multiple nucleoid copies per cell, and the aggregates localize to nucleoid-free areas in these cells[21]. To investigate the effect of aggregates in non-pole regions of the cell on the protein mobility, we treated cells with cephalexin, the antibiotic that prevents Z-ring constriction by inhibiting the

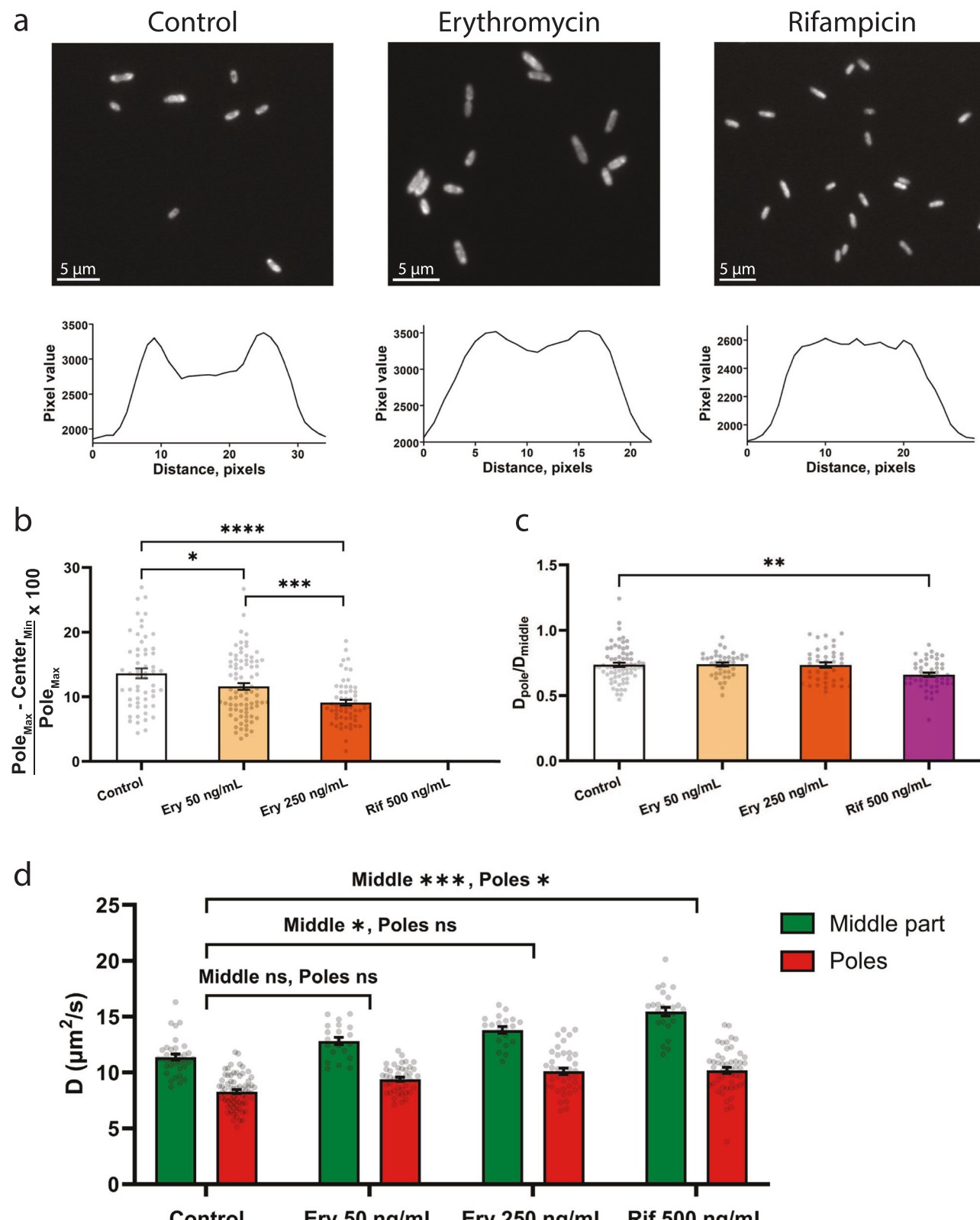

FtsI protein, and reconstructed diffusion maps[39]. We analyzed BW25113-*mEos3.2* cells after one to two compromised divisions, i.e. after 4 to 7 h of cephalexin treatment (Fig. 6a). As elongated cells cannot be perfectly aligned, we focused on the qualitative analysis of the diffusion and displacement maps. In cells after one division (4-h treatment) we observed decreased lateral diffusion in the septation area of elongated cells (Fig. 6b). In cells after

two divisions (7-h treatment) we still observe a decreased $D_L$ in the middle of elongated cells, most likely in between two nucleoids, but the area is increased. We also observe additional areas with slower lateral diffusion, i.e. at the two loci of second divisions. Heterogeneity in diffusion coefficients was found in all the observed cells (Fig. 6b). Also, in all cephalexin-treated cells we observed increased pole areas with decreased $D_L$. For all cells, areas with a

**Fig. 2 | Effect of ribosomal distribution on protein diffusion in *E. coli* cells.**
**a** Widefield fluorescent imaging of RbsB-mRuby3 localization in BW25113-*rpsB::mRuby3*. Left panel—untreated cells, middle panel—250 ng/mL of erythromycin (Ery), right panel—500 ng/mL of rifampicin (Rif). RbsB-mRuby3 fluorescence intensity profiles along the main axis of *E. coli* cells. Conditions the same as for the top panel. **b** The relative distributions of the ribosomes were quantified using the equation: $100 \times (Pole_{Max} - Center_{Min})/Pole_{Max}$, where $Pole_{Max}$ and $Center_{Min}$ are the maximal intensities at the poles and minimal intensities in the cell center, respectively. Number of analyzed cells is 28 for untreated cells, 41 for cells treated with 50 ng/mL erythromycin and 29 for cells treated with 250 ng/mL. **c** Pole/

middle ratios of lateral diffusion coefficients ($D_L$) for untreated and antibiotic-treated BW25113-*mEos3.2*. Number of analyzed cells is 30 for control, 21 for 50 ng/mL erythromycin, 20 for 250 ng/mL erythromycin and 25 for 500 ng/mL rifampicin treated cells. The correlation analysis of ribosomal distribution and $D_L$ decrease at the poles is shown in Fig. S2. **d** Absolute values of measured lateral diffusion coefficients in untreated and antibiotic-treated cells. Data presented as a mean value ± standard error of the mean (SEM) and corrected for solvent effect. Significance level is presented as asterisk signs: (ns) for $p > 0.05$, (*) for $p < 0.05$, (**) for $p < 0.01$, (***) for $p < 0.001$ and (****) for $p < 0.0001$.

**Fig. 3 | Effect of cell aging on protein diffusion.**
**a** Brightfield images of *E. coli* BW25113-*mEos3.2* in the terminal stages of division. **b** Upper panel: lateral diffusion coefficient measured in the middle part, new pole and old pole of dividing cells. Number of analyzed cells = 42. Lower panel: two-by-two contingency table of two parameter sets "old/new" and "fast/slow". *P* value obtained by Fisher's exact test. **c** Left panel: brightfield image of *E. coli* BW25113-*popZ::eGFP*. Right panel: widefield fluorescence image of PopZ-eGFP localization for the same field of view as shown on the left. **d** Upper panel: lateral diffusion coefficient measured in the middle part, new pole and old pole of *E. coli* BW25113-*popZ::eGFP*. Number of analyzed cells = 23. Lower panel: two-by-two contingency table of two parameter sets "old/new" and "fast/slow" as in panel B. p-value obtained by Fisher's exact test. **e** Pole/middle $D_L$ ratios measured for dividing *E. coli* BW25113-*popZ::eGFP*. **f** Summary table of measured absolute $D_L$ values and pole/middle ratios for dividing BW25113-*mEos3.2* and BW25113-*popZ::eGFP* cells. Data presented as a mean value ± standard error of the mean (SEM) and corrected for solvent effect. Significance level is presented as asterisk signs: (***) for $p < 0.001$ and (****) for $p < 0.0001$.

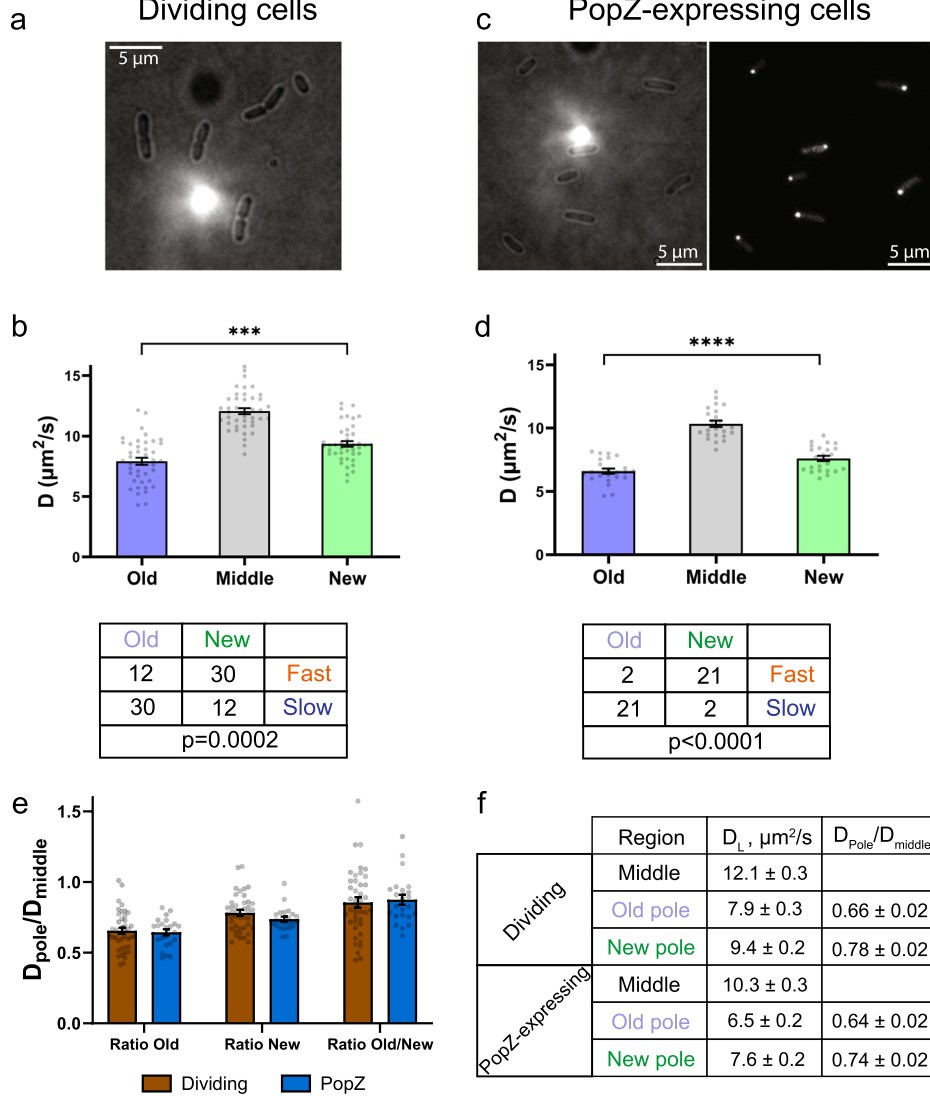

decreased diffusion coefficient correlate with a lower displacement density, which supports the idea that the slowed diffusion is due to an increased abundance of supramolecular assemblies and or aggregated proteins.

To increase the abundance of protein aggregates in division-compromised cells, we incubated cephalexin-treated cells for 1 h at 42 °C prior to SMdM. For the 4 h treatment we observed visual aggregates at the poles (and slowed diffusion) but not in the central part of elongated cells (Fig. S8a). The displacement maps of heat shocked and control cells are also similar, but the diffusion maps show a more pronounced decrease of $D_L$ at the division site (Fig. 6c). In heat-shocked cells treated with cephalexin for 7 h aggregates (Fig. S8b) were visible in between expected nucleoids[39]. Due to the stochastic nature of (protein) aggregate

formation, some cells had only one, other two or three optically-dense loci (Fig. S9). Overall, we find that areas with visual protein aggregates, triggered by the heat shock, had diminished displacement and localization density and slower diffusion than areas without these aggregates (Fig. 6c and Fig. S10). The decrease in the diffusion coefficient is comparable to that in the pole regions of heat-shocked cells without cephalexin treatment. Also, we observed a lower $D_L$ in regions close to the membrane, which is in agreement with the distribution of protein aggregates in division-inhibited cells[21]. In summary, we find a strong correlation between apparent slower lateral diffusion and the presence of protein aggregates, irrespective of whether they are formed at the poles or in middle regions of division compromised cells.

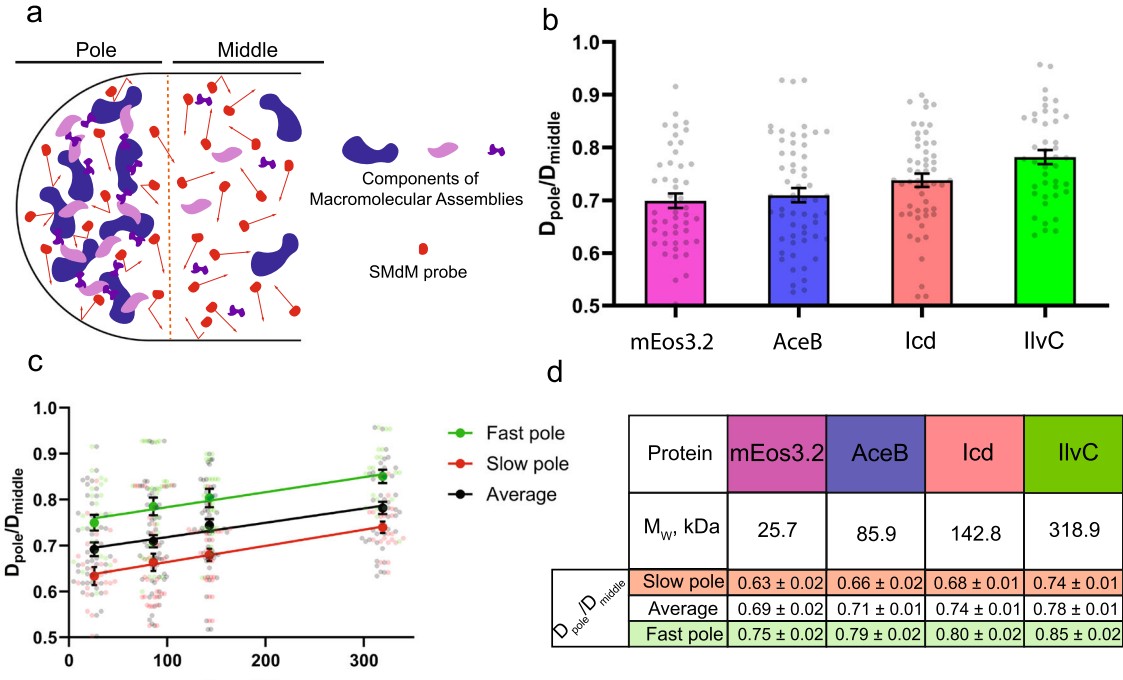

**Fig. 4 | Mass-dependent behavior of SMdM probe. a** Possible mechanism for reduced diffusion at the cell poles (see "Discussion"). **b** Pole/middle $D_L$ ratios for SMdM probes with different molecular masses in rifampicin-treated cells. Colors of bars are the same as in the Table of (**d**) (BW25113-*mEos3.2*—magenta), BW25113-*aceB::mEos3.2*—blue, BW25113-*icd::mEos3.2*—orange and BW25113-*ilvC::mEos3.2*—green. Number of cells is 24 for BW25113-*mEos3.2*, 29 for BW25113-*aceB::mEos3.2*, 27 for BW25113-*icd::mEos3.2* and 21 for BW25113-*ilvC::mEos3.2*.

**c** Pole/middle $D_L$ ratio as a function of SMdM probe $M_W$ for old and new poles separately. "Fast" and "slow" diffusion were used as proxy of old and new poles. Linear regression model fitting show $r^2 = 0.95$ for "fast" and $r^2 = 0.97$ for "slow" poles. The residuals plot can be found in supplementary Fig. S7. Data presented as a mean value ± standard error of the mean (SEM) and corrected for solvent effect. **d** Table of measured pole/middle $D_L$ ratios in *E. coli*, expressing mEos3.2 or mEos3.2 fusion proteins.

## Discussion

We have investigated the dynamic structure of the *E. coli* cytoplasm, using single-molecule displacement mapping (SMdM), to determine the spatial distribution of probe proteins and measure their lateral diffusion coefficient ($D_L$). This allowed us to determine the effect of the presence of polyribosomes and protein aggregates on protein diffusion in *E. coli*, and to connect heterogeneity in data density and protein diffusion to aging of the cells. In our previous work by Śmigiel et al. [1], we observed that cells have a slow and fast pole but did not explore the underlying mechanisms. In Mantovanelli et al. [2], we developed an alternative method for analysis of single-molecule displacement data and validated the software on antibiotic-treated and dividing cells, akin Figs. 2c, d, 3a, b. We now present comprehensive datasets with new experimental conditions to reveal the molecular basis for the spatial organization of the bacterial cytoplasm, and place the observations in a biologically relevant context.

To evaluate the effect of (poly)ribosomes on protein diffusion we treated *E. coli* with erythromycin and rifampicin. Erythromycin induced a change in abundance of ribosomes at the cell poles but had no significant effect on protein mobility, measured as pole/middle ratio of $D_L$. Rifampicin treatment dissipated the (poly)ribosome accumulation at the poles leading to more uniform distribution of the ribosomal subunits and diminishing the differences in volume exclusion between pole and middle areas of the cell [15]. We also show that rifampicin treatment increases the overall mobility of proteins in the cytoplasm (Fig. 2d), presumably by decreasing the effective viscosity of the cytoplasm as a result of mRNA depletion [36]. Moreover, upon rifampicin treatment the nucleoid expands and occupies the whole cytoplasmic volume of the cell, which leads to a lower compaction of the DNA and increased nucleoid mesh size [40]. Importantly, when we remove the (poly)ribosomes as possible obstruction factor for diffusion at the poles, we observe an unexpected decrease in pole/middle $D_L$ ratio (Fig. 2c). This decrease suggests

that obstruction factors other than (poly)ribosomes have a major effect on protein diffusion in the pole regions.

We show that two poles within an *E. coli* cell significantly differ in protein abundance and mobility (Fig. 3). This asymmetry in the properties of the pole regions can be explained by cell aging. In binary-dividing cells the distribution of misfolded and damaged proteins is asymmetric as the old pole accumulates more protein aggregates, which is an aspect of cell aging that leads to a lower viability [21]. We establish a significant correlation between pole age and the decrease in lateral diffusion coefficient. We see that "fast" and "slow" poles of *E. coli* correspond to new and old, respectively. We observe differences between the poles on the scale of one cell division, and it would have been advantageous if the analysis could have been extended over multiple generations, e.g. by using the "mother machine" of Suckjoon Jun et al. [41]. In the microfluidic "mother machine" device the mother cell is located in the dead end of a channel and it is evident from published data that the cells are not immobile, which is incompatible with SMdM as cells have to immobile for the duration of the measurements [41]. For these and other technical hurdles (consistent levels of expression) we did not pursue measurements over multiple generations.

Pole-organizing protein PopZ is a marker of the old pole and forms biomolecular condensates with liquid-liquid phase separation properties [7]. In our experiments, PopZ-condensates are not affecting pole/middle $D_L$ ratios (Fig. 3e). Thus, whatever the molecular nature of this condensate is, it is not affecting the mobility of mEos3.2 protein (25.7 kDa $M_W$).

Pole/middle $D_L$ ratios linearly increase with the molecular mass of the SMdM protein probe. We propose two mechanisms for this increase; both assume that diffusion at the poles is reduced by assemblies formed by protein aggregates [21], biomolecular condensates [11], and or plasmids [17] (Fig. 4a). First, aggregated macromolecules at the cell poles may shape structures, akin porous beads, that exclude larger proteins and allow small proteins to penetrate. Hence, the diffusion of small proteins will be relatively

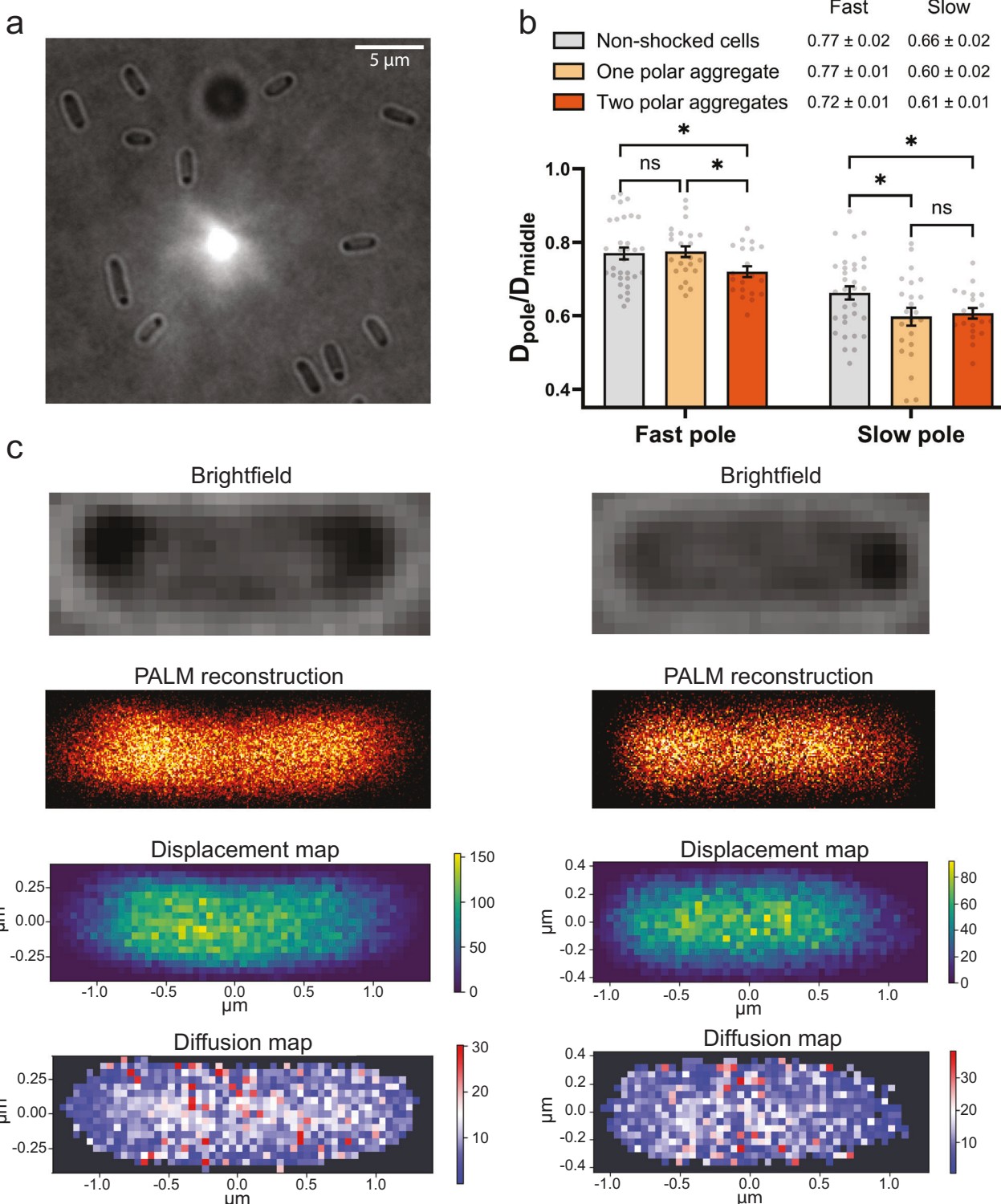

**Fig. 5 | Induced protein aggregation slows down protein diffusion in *E. coli*.**
**a** Brightfield images of *E. coli* BW25113-*mEos3.2* with heat-induced protein aggregation. **b** Pole/middle $D_L$ ratios for non-shocked and heat-shocked cells with one and two optically-dense aggregates. Data presented as mean ± standard error of the mean (SEM), the number of non-shocked and shocked cells with one- and two-pole aggregates was 32, 23 and 20, respectively. **c** Brightfield image, PALM reconstruction, displacement and diffusion maps for *E. coli* BW25113-*mEos3.2* with two (left) and one (right) pole with heat-induced aggregates. The pixel bin size of the displacement and diffusion maps is 50 nm. Color map for displacements represents the number of displacements per pixel. Diffusion maps were reconstructed by fitting displacements in each pixel bin with Eq. (3).

more affected, like they are in a size-exclusion chromatography experiment. Second, while the supramolecular complexes create excluded volume that slows down proteins of all sizes, such structures would have a smaller surface area than the individual molecules in the middle region of the cell. The

decrease of surface area reduces the added excluded volume for the larger proteins and, accordingly, reduces the friction that hinders their diffusion. These two mechanisms can coexist: the first one slows down smaller proteins at the cell poles and the second one reduces the friction experienced by

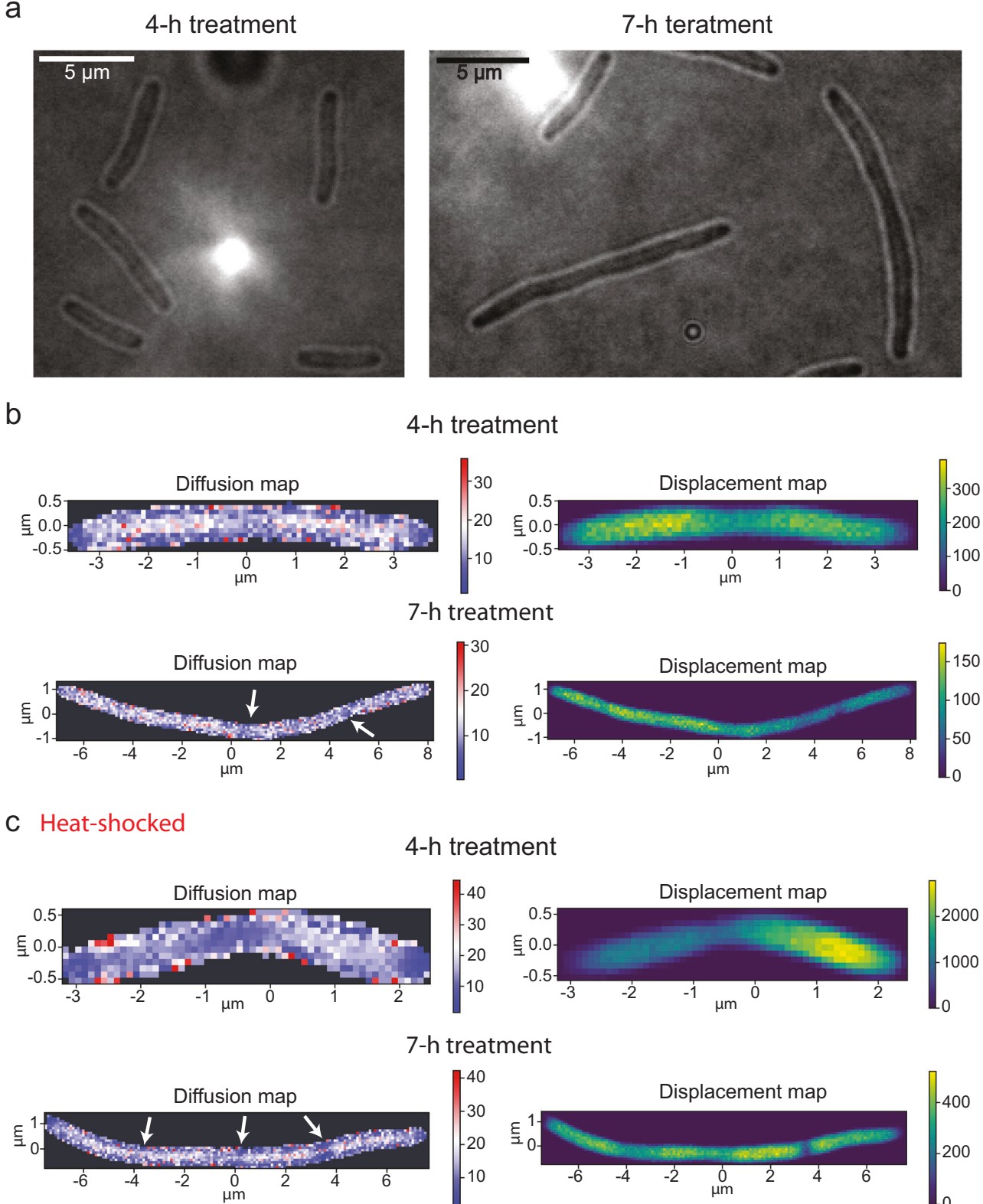

**Fig. 6 | Protein diffusion in division-inhibited E. coli cells. a** Brightfield images of *E. coli* BW25113-*mEos3.2* treated with 20 μg/mL cephalexin to inhibit cell division. 4-h treatment to observe cells with one compromised division, 7-h for two compromised divisions. **b** Diffusion and displacement maps of *E. coli* cells treated with cephalexin for 4 and 7 h. Pixel bin size is 100 nm and diffusion maps were reconstructed by fitting displacements in each pixel bin with Eq. (3). Arrows pointing to areas of lower protein mobility in cells treated with cephalexin for 7 h. **c** Diffusion and displacement maps of *E. coli* BW25113-*mEos3.2* cells treated with cephalexin for 4 and 7 h plus protein aggregation induced by heat-shock for 1 h after before SMdM. The pixel bin size of the displacement and diffusion maps is 100 nm. Color map for displacement maps represents the number of displacements per pixel. Diffusion map reconstructed by fitting displacements in each pixel bin with Eq. (3). Brightfield images of cells can be found in Fig. S8a (left panel, cell in the black box) for 4 h treatment with cephalexin and in Fig. S8b for 7 h treatment. Arrows point to areas of lower protein mobility in the 7 h treated cells.

larger proteins at the pole relative to the middle of the cell[42]. Both scenarios are in accordance with a mechanism in which damaged/misfolded proteins are passively translocated to the poles upon cell division[23] and a higher abundance of pole-localized supramolecular structures in the old pole. Indeed, we observe a lower protein probe density in the old pole than in the new pole of dividing and non-dividing cell (Fig. S3).

We show that the heat shock-induced protein aggregation decreases $D_L$ in the pole areas (Fig. 5). When optically-visible aggregates are present only in one pole, the pole/middle $D_L$ ratio decreases only in that pole (i.e. the slow or old pole). These aggregates thus add to the hindrance of diffusion by the non-visible obstruction factors. Our observations are in line with those of Coquel and Lindner, who showed that the majority of protein aggregates locate to the old pole of the bacterial cell[16,21]. It is likely that the heat shock-induced aggregation is not leading to new foci, but it may enlarge the nuclei of misfolded proteins already present in the cell.

*E. coli* cells treated with cephalexin show alternating areas of slow and fast lateral diffusion. Knowing the doubling time of the cells, we were able to track the elongation and to estimate the number and location of the compromised divisions. Analysis of the diffusion maps shows that the compromised division sites are overlapping with areas of decreased $D_L$ (Fig. 6, Fig. S10). Also, the data density in these regions is lower than in the nucleoid, implying the presence of obstructing structures and increased excluded volume. We find optically-visible aggregates at these sites when a mild heat shock is applied to cephalexin-treated cells and this reduces the mobility of proteins even further (Fig. S9). These aggregates seem to distribute over the nucleoid-free areas, as they are excluded from the meshwork of DNA[39]. The decrease of $D_L$ in areas of aggregation, not affected by confinement, favors the idea that diffusion limitation due to the geometry of the pole is small compared to the contribution from macromolecular aggregation.

In summary, we have used single-protein diffusion to characterize the dynamic structure of the bacterial cytoplasm under standard conditions of growth, heat shock stress, and antibiotic treatment. We analyzed the motion of the photoactivatable fluorescent protein mEos3.2, which has no known interaction or function in *E. coli*. We conclude that the localization of ribosomes and PopZ-dependent biomolecular condensates are not responsible for the slowed diffusion in the pole regions. We find that the reduced protein mobility at the cell poles is associated with local protein aggregation and the formation of diffusion obstruction factors. The slower diffusion at the old poles may negatively impact metabolic processes and may contribute to cell aging[25]. On the other hand, a slower diffusion may reduce the intramolecular collisions in the crowded cellular environment, which have been postulated to be detrimental for proper (re-)folding of proteins and protein-protein interactions[43]. We hypothesize that maintaining protein mobility within a narrow range, which is different for the poles and middle region of the cell, can be important for the balancing of reactions and interactions in the bacterial cell."

## Materials and methods
### Strains and plasmids used in this work
*E. coli* strain BW25113 [F-, Δ(araD-araB)567, ΔlacZ4787(::rrnB-3), λ-, rph-1, Δ(rhaD-rhaB)568, hsdR514] was used in this work. For storage and cloning we used *E.coli* strain DH5α [F-, Δ(argF-lac)169, φ80dlacZ58(M15), ΔphoA8, glnX44(AS), λ-, deoR481, rfbC1, gyrA96(NalR), recA1, endA1, thiE1, hsdR17]. Strains expressing mEos3.2 fluorescent protein and mEos3.2 fusions were taken from our previous work[1]. All these strains carry a pBAD vector with insertion of the gene of interest under the control of the arabinose promoter. As a source for cloning of the *popZ* gene we used a codon-optimized nucleotide sequence obtained from GeneArt Service (Thermo Fisher Scientific), and pAC06 Gap1Cterm-eGFP plasmid from our laboratory collection as a source of not codon-optimized sequence of eGFP gene. pZ8-Ptac vector coding PopZ-eGFP fusion protein under *Ptac* promoter was obtained by USER cloning and then transformed, using the heat shock method for chemical competent *E. coli* DH5α cells. DNA was then isolated via plasmid preparation, using the NucleoSpin Plasmid kit (MACHEREYNAGEL), and subsequently sequenced via Sanger sequencing by Eurofins Genomics. Next, the pZ8-Ptac_PopZ-eGFP vector was retransformed to BW25113 for SMdM measurements. The list of all primers can be found in the Supplementary Table S1. To observe the cellular location of ribosomes we used *E. coli* BW25113-*rpsB::mRuby3* with a chromosomal integration of mRuby3 fluorescent protein gene 3' of the gene for 30 S ribosomal protein S2. Table 1 lists all the *E. coli* strains and plasmids used in this work (see also[44,45]).

### Culturing conditions and cell growth
Lysogeny broth (LB) was prepared using standard recipe and sterilized by autoclaving. Mops-buffered minimal media (MBM) was prepared as described in[1,46]. All preculuring and culturing conditions for SMdM procedure were used as described in[1]. Briefly, cells were precultured overnight in 3 mL of LB media supplemented with the appropriate concentration of antibiotic(s) at 30 °C with shaking at 200 rpm, after which LB preculture was diluted 100-fold in 3 mL of MBM media supplemented with 0.1% (v/v) glycerol and antibiotic(s) and incubated overnight at 30 °C with shaking at 200 rpm. After overnight culturing the cells are still in the exponential growth phase (Fig. S11), that is, because the cells are grown at a relatively slow rate. On the next day, the MBM culture was diluted into fresh, prewarmed MBM with 0.1% (v/v) glycerol plus antibiotic(s) to a final $OD_{600}$ of 0.05 and grown for 5 h before microscopy at 30 °C with shaking at 200 rpm. For growth of the BW25113-*rpsB::mRuby3* strain no antibiotics were used.

To overexpress mEos3.2 and fusion constructs 0.1% (w/v) L-arabinose was used unless stated otherwise. mEos3.2 production was induced for 5 h and fusion constructs for 2 h. mEos3.2 induction in cephalexin-treated cell was done with 0.5% of L-arabinose. PopZ-eGFP fusion protein was expressed alongside with mEos3.2 protein by adding 0.5 mM Isopropyl ß-D-1-thiogalactopyranoside (IPTG) 1 h before microscopy. Concentrations of antibiotics used in this work: 100 μg/mL ampicillin dissolved in MQ water, 50 μg/mL kanamycin dissolved in MQ water, 50 or 250 μg/mL erythromycin dissolved in EtOH, 500 μg/mL rifampicin dissolved in DMSO, 20 μg/mL cephalexin dissolved in MQ water. If not used a selective marker, cells were treated with antibiotic for one hour, unless stated otherwise. For heat-shock treatment, the cells grown at 30 °C were transferred to incubator at 42 °C and shaking at 200 rpm for 1 h.

A 100-fold dilution of overnight LB culture was added to 100 μl of LB media in the wells of 96-well plate (Greiner Bio). 1 μl of overnight LB culture was added to wells. Optical density at 600 nm was measured every 10 min for 12 h in the SpectraMax® ABS plus (Molecular Devices) plate reader with orbital shaking at 200 rpm. To obtain specific growth rates the data was fitted with a logistic curve, and the doubling time was calculated as $ln(2)$ divided by the specific growth rate.

### Fluorescence microscopy
We have used a home-built inverted wide-field Olympus IX-81 microscope with TIRF objective with high numerical aperture (100x, 1.49 NA, Olympus) for all types of microscopies. Images were captured using EM-CCD camera (C9100-13, Hamamatsu). Brightfield images were captured without any EM gain for all measured cells. Florescence of rpsB-mRuby3 fusion protein was exited with a 561 nm laser (OBIS LS 561-150) with power set to 5 mW. We used highly inclined and laminated optical sheet (HILO) illumination to reduce the amount of background fluorescence and collected emitted light in the spectral range from 570 to 640 nm, using a ET 605/70 M bypass filter (Chroma). An acquisition time of 50 ms was used to accumulate sufficient signal. For localization microscopy of PopZ-eGFP fusion protein an excitation laser of 488 nm (OBIS LS 488-150) at 5 mW power was used to excite eGFP fluorescence in HILO mode. The emitted light was collected in the spectral range from 570 to 640 nm, using a ET 605/70 M bypass filter (Chroma) to spatially separate signals from eGFP and the green state of mEos3.2. The excitation efficiency of the fluorescent proteins was different at 488 nm (0.56 for green state of mEos3.2 and 1.0 for eGFP due to fpbase.com) and thus at longer wavelength the relative difference in emitted signal is higher.

## Table 1 | List of *E. coli* strains and plasmids used in this work

| Bacterial strain | Used for | Source/Reference |
|---|---|---|
| BW25113 | Cloning of genes of interest for SMdM | 44 |
| DH5α | Storage of plasmids an initial cloning | 45 |
| BW25113-*mEos3.2* | Overexpression of mEos3.2 florescent protein for SMdM measurements | 1 |
| BW25113-*rpsB::mRuby3* | Localization microscopy of ribosomal density in bacterial cells | Gift from System Biology Group of University of Groningen |
| BW25113-*popZ::eGFP* | Localization microscopy of PopZ, determining the old bacterial pole | This work |
| BW25113-*aceB::mEos3.2* | Overexpression of AceB-mEos3.2 fusion for SMdM measurements | 1 |
| BW25113-*icd::mEos3.2* | Overexpression of Icd-mEos3.2 fusion for SMdM measurements | 1 |
| BW25113-*ilvC::mEos3.2* | Overexpression of IlvC-mEos3.2 fusion for SMdM measurements | 1 |
| **Plasmid** | | |
| pBAD_MGGTGGS-mEos3.2-6his | Coding mEos3.2 protein | 1 |
| pBAD_MGGTGGS-aceB-mEos3.2-6his | Coding AceB-mEos3.2 fusion protein | 1 |
| pBAD_MGGTGGS-icd-mEos3.2-6his | Coding Icd-mEos3.2 fusion protein | 1 |
| pBAD_MGGTGGS-ilvC-mEos3.2-6his | Coding IlvC-mEos3.2 fusion protein | 1 |
| pZ8-Ptac | Vector with IPTG inducible promoter | Addgene |
| pMA-RQ-PopZ_linker_mRuby | Source of *popZ* gene | GeneArt |
| pAC06_Gap1Cterm-eGFP | Source of *eGFP* gene | Collection of Membrane Enzymology Group |
| pZ8-Ptac_PopZ-eGFP | Coding PopZ-eGFP fusion protein | This work |

## Super-resolution diffusion measurements

**Data acquisition**. Detailed protocol can be found in[1,47]. Briefly, diluted to $OD_{600}$ 0.05 overnight MBM preculture (see Culturing conditions paragraph) was grown for 5 h at 30 °C with shaking at 200 rpm until culture density of ~0.15. Overexpression of mEos3.2 or fusion proteins was done as mentioned in Culturing conditions paragraph. After that 500 μl of culture in spun down and resuspended in 150 μl of remaining media. 3 μl of bacterial cell culture are put on cleaned by sonication in 5 M KOH 1.5H high-precision glass slides (170 μm thickness, Carl Roth GmbH & Co KG). To prevent cells from moving while measurement and to keep them moistened we put agarose pad on top of droplet of cell culture. That agarose pad was by mixing two times concentrated MBM media, 1.5% melted agarose in MQ water inside a 8-mm-diameter hole in PDMS chamber. Glycerol was added to pads at final concentration of 0.1% to match the composition of the growth MBM media.

Selected 250×250 pixels fields of view were illuminated with temporally controlled 405 and 561 nm laser pulses as described in[1] and[47]. In short, 405 nm laser pulse (OBIS 405 LX, 50 mW max. power) was applied to photoconvert mEos3.2 from green florescence state (507 nm ex. / 516 nm em.) to red (572 nm ex. / 580 nm em.) and two readout beams of 561 nm laser (OBIS LS 561-150) were applied with time separation (Δt) of 1.5 ms. Fluorescent signal was collected by a EM-CCD camera (C9100-13, Hamamatsu), using a ET 605/70 M bypass filter (Chroma). Several movies were captured per field of view, resulting in a total number of 100,000 frames per field of view, which were saved as .stk files (MetaMorph stack, Molecular Devices). The total measurement time for one field of view is approximately 30 to 40 mins. The autofocus function of the microscope was enabled to avoid z-drift. Within this time range, the cells were relatively immobile. Separate movies were concatenated using a python script and then converted to .tif file. As for further analysis the correct order of odd and even frames is essential, we used information of background fluorescence intensity from the first 10 frames of each movie to verify the correct frame order, because it is different for odd and even frames.

**Data analysis**. Fluorescent peaks were detected using STORM-analysis package developed by the Zhuang lab (http://zhuang.harvard.edu/software.html) without considering z-dimension as we are analyzing 2-dimentions motion. Coordinates of detected pecks were xy-drift corrected and saved as a .hdf5 file. The obtained xy coordinates of fluorescent peaks were clustered based on the densities of point clouds, using the Voronoi tessellation method. For analysis of dividing cells, if daughter cells were clustered together, this cluster was split in two and analyzed separately. After that, detected point clouds were rotated along long axes, determined by first eigenvectors of point cloud covariance matrix, to facilitate the pixilation of further reconstruction of diffusion maps.

Two-dimensional displacements of protein were detected as a Euclidean distance between peak at odd frame and consecutive even frame at fixed time interval between two readout laser beams. Maximum distance between pairs was set to 600 nm to reduce the amount of ambiguity[1,47]. To evaluate the lateral diffusion coefficient ($D_L$), probability density distribution of measured displacements as a function of time separation (Δt) is fitted with adjusted probability density function (PDF) of a 2-dimensional random-walk diffusion model:

$$p_0(r, \Delta t) = \frac{2r}{4D_L \Delta t} e^{-\frac{r^2}{4D_L \Delta t}} \tag{1}$$

Where $D_L$ is a lateral diffusion coefficient, r is peak-to-peak displacement and Δt is time separation between 561 nm readout laser pulses. This $p_0(r, \Delta t)$ describes Rayleigh distribution. To compensate for ambiguous peak pairing, we introduce a linear correction factor, relying on the assumption that detected "background" peaks are evenly distributed within field of view. So now probability density function transforms to:

$$p_{corr}(r, \Delta t) = \frac{2r}{4D_L \Delta t} e^{-\frac{r^2}{4D_L \Delta t}} + br \tag{2}$$

Where b is a background correction coefficient. But as we also restrict the maximum displacement, we have to normalize total PDF to have the integral of it to be equal to 1. That is done by dividing Eq. 2 by integral from 0 to maximum search radius ($r_{max}$) of $p_{corr}(r, \Delta t)$:

$$p(r, \Delta t) = \frac{1}{1 - e^{-\frac{r_{max}^2}{4D_L \Delta t}} + \frac{b}{2} r_2^{max}} \left( \frac{2r}{4D_L \Delta t} e^{-\frac{r^2}{4D_L \Delta t}} + br \right) \tag{3}$$

We used Eq. 3 to fit the displacements using maximum likelihood estimation (MLE) using $D_L$ and b as fitting parameters to get the values of lateral diffusion coefficient. Based on the average dimensions of the cells, we used 20% of the total length of the (rotated) cells as pole region, and the remaining 60% as the middle part. DAPI staining of *E. coli* cells showed that the nucleoid-occupied area largely overlaps with the 60% middle part (Fig. S12). To reconstruct diffusion maps we binned each cell into square selections with a side of 100 nm and fitted displacements starting within a bin with Eq. 3.

For calculating $D_L$ we selected cells with the number of observed displacements between 1000 and 20.000. A lower amount of data results in bias fitting and in a relatively large influence of background noise and ambiguous pairing. For higher resolution of reconstructed diffusion maps, we were not following this restriction for the whole cell (or for central or pole regions), but we used lower and upper boundaries of 10 and 20.000 for each bin separately.

### Super-resolution image reconstruction
To make representative PALM reconstructions we used the QuickPALM plugin (https://imagej.net/plugins/quickpalm) for ImageJ with default settings. Fluorescent peaks detected on all frames acquired in the SMdM measurements were used to obtain this reconstruction with 10x magnification.

### Statistical analysis
We used the Shapiro-Wilk normality test to evaluate normality of data distribution. The null hypothesis for normal distribution was accepted if $p > 0.01$, as we set a significance level of 1%. Depending on the results of the normality test we applied the two-side Student T-test or Mann-Whitney U-test to compare mean values of two experimental samples both with significance level of 5 percent.

Two-by-two contingency tables were analyzed using Fisher's exact test with the significance level of 5 percent to evaluate the correlation between two pairs of parameters.

We fitted a line equation, using the linear regression model as described in the paragraph "Mass-dependent behavior of SMdM probe" to evaluate the parameters of the line equation. The evaluated parameters were used for extrapolating the observed behavior towards higher $M_W$ values. The analysis of the residuals was performed by plotting the observed value minus the value predicted by the regression model. Residuals higher than zero indicate an underestimation of the fit, while residuals lower than zero indicate an overestimation.

### Reporting summary
Further information on research design is available in the Nature Portfolio Reporting Summary linked to this article.

### Data availability
The source data behind the graphs in the paper can be found in Supplementary Data 1. The raw microscopy data is available from the corresponding author upon request.

### Code availability
Developed code for modulating laser pulses using PCI-6602 programmable card (National Instruments) and for SMdM analysis is available on the Github repository of Membrane Enzymology Laboratory: https://github.com/MembraneEnzymology/smdm/tree/main/Microscopy[48].

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

## Acknowledgements
We thank Dr. Wojciech M. Śmigiel and Luca Montavanelli for their contribution in setting up the SMdM microscopy and making engineered *E. coli* strains available. We thank José Vila Chã Losa and Prof. Matthias Heinemann for a gift of *E. coli* strain BW25113 *rpsB::mRuby3*. The research was funded by NWO National Science Program "The limits to growth" (grant number NWA.1292.19.170).

## Author contributions
Dmitrii Linnik—Conceptualization, experimental design, cloning, microbiology, data acquisition, data analysis, Python scripting, statistical analysis, writing of original draft. Ivan Maslov.—Data analysis, writing, reviewing and editing. Christiaan Michiel Punter—Data analysis, Python scripting, IT supervision. Bert Poolman—Conceptualization, experimental design, project supervision, writing, review and editing.

## Competing interests
The authors declare no competing interests.
