## [Peer Review File · Communications Biology]

Reviewers' comments:

Reviewer #1 (Remarks to the Author):

This study uses a new single-molecule technique to probe the diffusion of a fluorescent protein in the E.coli cytoplasm at high temporal and spatial resolution. The results provide interesting insights into the dynamics of the cytoplasm, and reasons for faster and slower diffusion in different regions. The experiments are well-designed and well-presented, and I have only minor comments:

1. Introduction line 36: "The biochemical reactions are taking place in uninterrupted cytoplasm...".

This sentence is rather clumsy and it's not immediately obvious what it means. Maybe something like "Most biochemical reactions take place in a continuous cytoplasm without membrane barriers..."

2. Intro lines 76-83. The authors seem to be suggesting that heterogeneity in membrane lipid composition arises because of the site of synthesis of the lipids. That idea doesn't work because lipids in the E. coli plasma membrane can diffuse and mix on much faster timescales than the cell cycle. There must be specific factors that recruit a different lipid mixture to the poles as compared to the lateral membrane - membrane curvature is a strong possibility, but protein factors could also be involved.

3. Results line 124: more precise than "...is not a major cause for lower mobility..." would be "...is not the only cause for lower mobility.."

4. Results lines 125-126: "We attribute the overall increase in protein mobility to a decreased cytoplasmic viscosity". This is a completely empty statement, unless the authors are trying to suggest that there is a decrease in the actual cytoplasmic viscosity, as opposed to the effective viscosity. I don't think they are trying to say that. It would be better to give here the explanation that they give in the discussion (line 247) that the effective viscosity decreases because of mRNA depletion.

5. Line 246: should be "...by decreasing the effective viscosity.." not increasing!

Reviewer #2 (Remarks to the Author):

The paper deals with the mechanisms underlying non-homogenous diffusion of proteins within bacterial cells, as recently studied by the authors.

The authors test two hypotheses:

first, that since large polysomes can't enter the mid-region of the cell containing the nucleus, the observed diffusion constant in mid cell would be lower. Using two antibiotic perturbations they show that this is not consistent with the data. Second, they show, both relying on wild-type observations and on

perturbations, that the data is consistent with a picture where protein aggregates (known to accumulate at the older poles)

lead to the observed lower diffusion.

I found the paper very well written and the evidence provided compelling and the results interesting, and therefore recommend the paper for publication.

Some minor comments:

- It would be good to elaborate on the experimental methods earlier in the paper, rather than leave it for the discussion.

- Related to that, a large part of the discussion reads more like a summary - it would have been better to instead elaborate on the significance of the results - why is the inhomogeneous diffusion important? How does it

contribute to/affect cell aging, if at all?

- In Eqs. 1-3, a factor of pi seems to be missing in the probability distribution. Does this affect any of the results?

Reviewer #3 (Remarks to the Author):

The manuscript is clearly written, and provides statistically significant and convincing results about the effect of protein aggregation, and further processes as, notably, cell aging, on the diffusion behaviour of mEos2 and protein fusions of different molecular weight, in different regions of bacterial cells. As the entire manuscript relies on the relatively novel SMdM approach, I think providing in the figures some more direct examples of such measurements analysis (as in Figure S4) would be beneficial for clarity. Moreover, a small number of further controls and possibly some FRAP measurements could make the manuscript even more solid.

General observations:

1) All measurements are acquired with SMdM which is defined by the authors themselves as a „novel technique“, although plenty of references are provided, with the purpose to reach more readers, it would be useful to add few lines to introduce the technique and its advantages in this particular application over spt-PALM.

2) In many of the figures, the authors look at the diffusion in the middle and in the poles of cells displaying statistically significant differences. It would be interesting to show how diffusion changes over cell length, I am not sure how many independent points could be provided in terms of spatial resolution, I am wondering if the data could be used to see if the transition in diffusion rates between middle and poles is sharp or relatively gradual. In this context, it would be possibly relevant to keep into account the nucleoid localization more precisely (see also point (12)).

Detailed questions and observations:

3) L45 We now determined the molecular basis for the differences in mobility of proteins in the different regions of the cell.

While I share the spirit of this statement, I feel it's too strong of a claim for the data provided in the manuscript, perhaps the authors could mild it down a little?

4) Fig 2B-2D

To further substantiate the claim in line 123 could the authors verify if $\Delta(D)$ between middle and poles correlates (or not) with the peak to valley ratio in the fluorescence intensity profile for a given set of samples (control, Ery 50 ng/mL and so on) providing a plot of $\Delta(D)$ vs $100 \times (PoleMax - CenterMin) / PoleMax$

5) L139 We also observed an unequal distribution of mEos3.2 localizations in the cell: the central part of the cell (60% of the volume) had 78% of the displacements and the new and old poles (each 20% 140 of the volume) had 14% and 8% of the displacements, respectively.

Could the authors provide exemplary fluorescence images for mEos3.2 and analysis of fluorescence intensity along profile lines (similarly to Figure 2A). This would be important to see whether the difference in displacements correlates with the difference in concentration.

Could it be possible that the number of the displacements in the middle and/or in the poles is under-

(or over-)estimated using the 1.5 ms time step (time separation)? Would a different time step give a different ratio? Could the time step be varied in a small subset of samples? Given the difference between poles and middle of the cell is clearly statistically significant but still relatively small it would be interesting to see if varying such parameter could give a partially different outcome.

6) L155 We propose that the decrease of DL at the E. coli poles and differences between old and new pole of the cell are caused by protein aggregates, which accumulate more at the old than new pole.

Figure S4 shows the representative probability density function fitting profiles of the displacements in the middle part, it would be important to have similar examples for the poles, at least one for each condition of Figure 3.

Following up on point (5), is SMDM, for a given cell, showing single peak probability density functions of the diffusion at the poles and in the middle (distribution with a single peak like in Figure 1F) or are we in presence of higher heterogeneity in the distributions at the poles? In this second case, would it be possible some of the proteins at the poles are immobile and some other moves at a similar rate than the ones in the cell middle? Could a FRAP measurement (more crude but still effective) assess the eventual presence of an immobile fraction or better help in excluding its presence at the poles (this comment would be also important for the data in Figure 5)?

7) Figure 5. Could the authors add also representative fluorescence images under the brightfield ones?

8) Figure 6: Overall, we find that areas with visual protein aggregates, triggered by the heat shock, had diminished data density and slower diffusion than areas without these aggregates (Fig. 6C and Fig. S8). The decrease in the diffusion coefficient is comparable to that in the pole regions of heat-shocked cells without cephalixin treatment.

Could a fluorescence image be included in this figure and is it possible that acquiring with a different time step would change such percentages (Referring to "diminished data density")?

9) L229:

What is the actual spatial resolution in this case? Could the authors provide some kind of estimate? FCS (which, unlike FRAP, also belongs to the single molecule techniques family, this should be changed in the text) spatial resolution, by the way, is dictated by the confocal volume so it could also discriminate poles from middle.

10) L263 "However SMdM requires data accumulation for approximately 30 to 40 min during which the cell should be immobile".

As this number is one-two order of magnitudes higher than the typical acquisition times for FCS or even FRAP measurements, could the author reassure me and the potential readers that both the middle part and the poles were immobile within this time range?

While the agarose pads would probably keep cells rather stable, I am worried that for very elongated cephalixin-treated cells one could see some wobbling of the poles in such a long time range. Could the authors at least include a couple of time-lapse movies in the supplementary material to show cells (especially elongated cells) are indeed immobile both on the xy plane and in the focal direction in such time range?

Also, I feel that the information that one single measurement requires 30-40 minutes should be provided in the material and methods in case colleagues would want to perform similar experiments, as this could be a bottleneck for some applications.

11) L436 As a selection of pole regions of bacterial cells, we used 20 % of total length of the (rotated) cells, and the remaining 60% as middle part where (most of) the nucleoid localizes. DAPI staining of E. coli cells showed that the nucleoid area is approximately 60 % of the cell length, that is, in cells

growing exponentially on MBM supplemented with 0.1% glycerol (Fig. S10).

Bakshi et al. in 2014 showed a significantly smaller staining of the nucleoid using Sytox Orange, perhaps it would be good to rather base the distinction between nucleoid and poles on such dye localization, possibly acquiring also images with Sytox Orange staining?

Minor Observation:

In the supplementary material, the authors write "Brightfield" images, in the main text they write sometimes "Widefield" (but "Brightfield" directly in the figures), and differentiate fluorescence images calling them "Widefield Fluorescence" or just "Fluorescence", I would advise for higher consistency to av

Response to reviewer comments:

Reviewer #1 (Remarks to the Author):

This study uses a new single-molecule technique to probe the diffusion of a fluorescent protein in the E. coli cytoplasm at high temporal and spatial resolution. The results provide interesting insights into the dynamics of the cytoplasm, and reasons for faster and slower diffusion in different regions. The experiments are well-designed and well-presented, and I have only minor comments:

We thank the reviewer for their evaluation and comments on our manuscript.

1. Introduction line 36: "The biochemical reactions are taking place in uninterrupted cytoplasm...". This sentence is rather clumsy and it's not immediately obvious what it means. Maybe something like "Most biochemical reactions take place in a continuous cytoplasm without membrane barriers..."

For clarity, we revised the sentence as follows (see lines 36-37):

"Most of the biochemical reactions take place in a continuous cytoplasm, lacking membrane sub-compartments, and depend on Brownian motion for the biomolecules to interact"

2. Intro lines 76-83. The authors seem to be suggesting that heterogeneity in membrane lipid composition arises because of the site of synthesis of the lipids. That idea doesn't work because lipids in the E. coli plasma membrane can diffuse and mix on much faster timescales than the cell cycle. There must be specific factors that recruit a different lipid mixture to the poles as compared to the lateral membrane - membrane curvature is a strong possibility, but protein factors could also be involved.

We thank the reviewer for this comment. It is true that lateral diffusion of lipids in bacterial plasma membranes is mixing the lipid content of the plasma membrane.

We rewrote the text (see lines 78-85):

"Peptidoglycan synthesis takes place in the middle part of the cell, from where the "old" cell wall is continuously pushed toward the poles^{26,27}. This pole-oriented motion of the peptidoglycan guides the directed motion of membrane proteins¹⁴. Moreover, the altered curvature of the membrane at the cell poles may cause the accumulation of specific proteins and lipids at these sites^{28,29}. Hence, the content of the pole membrane differs from the rest of the cell, and the presence of specific factors in the membranes at the poles can act as a driving force for the formation of cytosolic pole-localized protein pools."

3. Results line 124: more precise than "...is not a major cause for lower mobility..." would be "...is not the only cause for lower mobility.."

We edited the text as suggested (see lines 134-138).

4. Results lines 125-126: "We attribute the overall increase in protein mobility to a decreased cytoplasmic viscosity". This is a completely empty statement, unless the

authors are trying to suggest that there is a decrease in the actual cytoplasmic viscosity, as opposed to the effective viscosity. I don't think they are trying to say that. It would be better to give here the explanation that they give in the discussion (line 247) that the effective viscosity decreases because of mRNA depletion.

This is a fair point. The revised sentence reads as follows (see lines 137 - 138):

“We attribute the overall increase in protein mobility to a decreased effective viscosity of the cytoplasm as a result of mRNA depletion upon rifampicin treatment”

5. Line 246: should be "...by decreasing the effective viscosity.." not increasing!

We edited the text as suggested by the Reviewer.

Reviewer #2 (Remarks to the Author):

The paper deals with the mechanisms underlying non-homogenous diffusion of proteins within bacterial cells, as recently studied by the authors.

The authors test two hypotheses:

first, that since large polysomes can't enter the mid-region of the cell containing the nucleus, the observed diffusion constant in mid cell would be lower. Using two antibiotic perturbations they show that this is not consistent with the data. Second, they show, both relying on wild-type observations and on perturbations, that the data is consistent with a picture where protein aggregates (known to accumulate at the older poles) lead to the observed lower diffusion.

I found the paper very well written and the evidence provided compelling and the results interesting, and therefore recommend the paper for publication.

We thank the reviewer for their evaluation and comments on our manuscript.

Some minor comments:

1) It would be good to elaborate on the experimental methods earlier in the paper, rather than leave it for the discussion.

We elaborated on the description of the technique in the Introduction, see lines (89-98):

“We determine protein mobility in *E. coli* using single-molecule displacement mapping (SMdM)^{1,33}. This recently developed technique uses photoactivated localization microscopy (PALM) for the localization of individual fluorescently-labeled proteins on consecutive frames of microscopy recording, while they diffuse inside a cell (Fig. 1D-F). Unlike conventional single-particle tracking PALM (spt-PALM), SMdM employs stroboscopic illumination to reduce the motion blur typical of spt-PALM data: The method ensures high spatial resolution of the diffusion mapping. In contrast to for instance, fluorescence correlation spectroscopy (FCS) and fluorescence recovery after photobleaching (FRAP), SMdM allows simultaneous acquisition of data in various regions of a cell or several adjacent cells, which allows the detection of a large number of molecules and thus enhances the precision and throughput of the diffusion measurements.”

2) Related to that, a large part of the discussion reads more like a summary - it would have been better to instead elaborate on the significance of the results - why is the inhomogeneous diffusion important? How does it contribute to/affect cell aging, if at all?

We rewrote the discussion and added the following (lines 329-326):

- 1) “We find that the reduced protein mobility at the cell poles is associated with local protein aggregation and the formation of diffusion obstruction factors. The slower diffusion at the old poles may negatively impact metabolic processes and may contribute to cell aging²⁵. On the other hand, a slower diffusion may reduce the intramolecular collisions in the crowded cellular

environment, which have been postulated to be detrimental for proper (re-)folding of proteins and protein-protein interactions⁴³. We hypothesize that maintaining protein mobility within a narrow range, which is different for the poles and middle region of the cell, can be important for the balancing of reactions and interactions in the bacterial cell.”

3) In Eqs. 1-3, a factor of pi seems to be missing in the probability distribution. Does this affect any of the results?

The equation is correct. The probability density of finding a particle at a specific x, y position, assuming it starts from the origin, is given by:

$$p(x, y) = \frac{1}{4\pi D \Delta t} \exp\left(-\frac{x^2 + y^2}{4D\Delta t}\right)$$

where D is the diffusion coefficient and Δt is the time interval.

To find the probability density at a certain distance r , or displacement, we have to consider all possible x, y coordinates that lay on a circle with radius r centered around the starting position of the particle. This means we have to integrate the aforementioned probability density function over all x, y positions on the circumference of the circle. This entails multiplying the probability density function by $2\pi r$, the circumference, which leads to the probability density function that we use for fitting the displacement data:

$$p(r) = \frac{2\pi r}{4\pi D \Delta t} \exp\left(-\frac{r^2}{4D\Delta t}\right) = \frac{r}{2D\Delta t} \exp\left(-\frac{r^2}{4D\Delta t}\right)$$

The same equation was used in the original SMdM paper: see Equation 1 in¹.

¹ Limin Xiang et al., “Single-Molecule Displacement Mapping Unveils Nanoscale Heterogeneities in Intracellular Diffusivity,” *Nature Methods* 17, no. 5 (May 2020): 524–30, <https://doi.org/10.1038/s41592-020-0793-0>.

Reviewer #3 (Remarks to the Author):

The manuscript is clearly written, and provides statistically significant and convincing results about the effect of protein aggregation, and further processes as, notably, cell aging, on the diffusion behaviour of mEOS2 and protein fusions of different molecular weight, in different regions of bacterial cells. As the entire manuscript relies on the relatively novel SMdM approach, I think providing in the figures some more direct examples of such measurements analysis (as in Figure S4) would be beneficial for clarity. Moreover, a small number of further controls and possibly some FRAP measurements could make the manuscript even more solid.

We thank the reviewer for their suggestions.

General observations:

1) All measurements are acquired with SMdM which is defined by the authors themselves as a „novel technique“, although plenty of references are provided, with the purpose to reach more readers, it would be useful to add few lines to introduce the technique and its advantages in this particular application over spt-PALM.

We now present a description of the SMdM method in the Introduction (see lines 89-98):

See our response to comment 1 of reviewer 2.

2) In many of the figures, the authors look at the diffusion in the middle and in the poles of cells displaying statistically significant differences. It would be interesting to show how diffusion changes over cell length, I am not sure how many independent points could be provided in terms of spatial resolution, I am wondering if the data could be used to see if the transition in diffusion rates between middle and poles is sharp or relatively gradual. In this context, it would be possibly relevant to keep into account the nucleoid localization more precisely (see also point (12)).

The reconstructed diffusion maps of Fig. 5C and 6B-C already show a relatively gradual change of the diffusion coefficient along the cell length. Below, we also show a dataset from a previous paper, where the diffusion maps are plotted for different pixel sizes. Thus, the information is already visible in the presented data.

To quantify the difference in the diffusion between the middle and the pole regions of the cells, we used a volume for each pole of 20%. This estimate is based on the geometry of the average cell in the dataset, i.e. the radius of the cell is approximately 20% of the cell length. As an additional check, we now show by DAPI staining that the middle region of the cell largely overlaps with the nucleoid.

Detailed questions and observations:

3) L45 We now determined the molecular basis for the differences in mobility of proteins in the different regions of the cell.

While I share the spirit of this statement, I feel it's too strong of a claim for the data provided in the manuscript, perhaps the authors could mild it down a little?

We rephrased the sentence (lines 45 - 47):

“In this work we investigate the molecular mechanisms underlying the differences in the mobility of proteins present in different regions of the cell.”

² Wojciech M. Śmigiel et al., “Protein Diffusion in *Escherichia Coli* Cytoplasm Scales with the Mass of the Complexes and Is Location Dependent,” *Science Advances* 8, no. 32 (August 12, 2022): eabo5387, <https://doi.org/10.1126/sciadv.abo5387>.

4) Fig 2B-2D

To further substantiate the claim in line 123 could the authors verify if $\Delta(D)$ between middle and poles correlates (or not) with the peak to valley ratio in the fluorescence intensity profile for a given set of samples (control, Ery 50 ng/mL and so on) providing a plot of $\Delta(D)$ vs $100 \times (PoleMax - CenterMin) / PoleMax$.

In agreement with our claim, we observe no significant correlation between the $100 \times (PoleMax - CenterMin) / PoleMax$ and either $\Delta(D)$ or the D_{pole}/D_{middle} ratio; the p-values of Spearman correlation are $p=0.33$ and $p=1.00$, respectively. We now include the figure below as Supplementary Figure S2.

Fig. S2. The ribosomal distribution profile does not correlate with the difference between diffusion coefficient at the center and poles of erythromycin-treated *E. coli*. The dependence of $100 \times (PoleMax - CenterMin) / PoleMax$ on the ratio of pole/middle lateral diffusion (A) and on the difference between D_{pole} and D_{middle} (B) are not significant; the Spearman's rank correlation coefficient, p-values are 0.33 for (A) and 1.00 for (B).

We mention this analysis in lines 133-138 of the main text and in the caption of the Figure 2C (line 655-656):

“Taken together these data indicate that the presence of polysomes at the cell poles is not a major cause for the lower mobility, because the difference in pole/middle D_L ratio is also found in cells where the ribosome subunits are evenly distributed; The pole/middle D_L ratio and the difference between D_L in the middle and at the poles do not correlate with the distribution of ribosomes (Fig. S2)”. We attribute the overall increase in protein mobility to a decreased effective viscosity of the cytoplasm as a result of mRNA depletion upon rifampicin treatment.

“The correlation analysis of ribosomal distribution and D_L decrease at the poles is shown in Fig. S2.”

5) L139 We also observed an unequal distribution of mEos3.2 localizations in the

cell: the central part of the cell (60% of the volume) had 78% of the displacements and the new and old poles (each 20% 140 of the volume) had 14% and 8% of the displacements, respectively.

Could the authors provide exemplary fluorescence images for mEos3.2 and analysis of fluorescence intensity along profile lines (similarly to Figure 2A). This would be important to see whether the difference in displacements correlates with the difference in concentration.

Could it be possible that the number of the displacements in the middle and/or in the poles is under- (or over-)estimated using the 1.5 ms time step (time separation)?

Would a different time step give a different ratio? Could the time step be varied in a small subset of samples? Given the difference between poles and middle of the cell is clearly statistically significant but still relatively small it would be interesting to see if varying such parameter could give a partially different outcome.

To minimize the photobleaching before the SMdM measurements, we did not record images of the non-converted green-fluorescent state of mEos3.2. To address the reviewer's concern, we analyzed the distribution of all fluorescent localizations of the SMdM experiment, independent of whether they formed a displacement pair (Figure S3). Figure S3 shows an unequal distribution of the localizations and displacements, with the least data available for the old pole.

Fig. S3. Data distribution of mEos3.2 displacements and localizations in *E. coli* BW25113-mEos3.2 cells; the data are presented as percentage of all observed displacements or localizations. The data distribution is significantly different for the old and new pole of dividing cells. Data presented as mean \pm SEM; the number of dividing BW25113-mEos3.2 cells is 43. The significance level is presented as asterisk signs: (****) for $p < 0.0001$.

Adjusting the time step does not impact the measured distribution of the localizations, as it solely alters the temporal resolution of observed displacements, without affecting the overall spatial resolution.

To limit the effects of confinement on the apparent diffusion coefficients we use the shortest possible time interval, Δt , so that the expected mean displacement is smaller than the cell width. The shortest time interval possible with our microscopy setup is 1.5ms. Computer simulations of particle diffusion in a rod-shape cells, presented in our previous studies³ and³, show that the confinement effect leads to underestimation of the diffusion coefficients. For particles with a diffusion coefficient $D_L=10 \mu\text{m}^2/\text{s}$, the simulations suggest 15-20 % error in the apparent diffusion coefficient; longer Δt aggravates the error as shown in Fig. R1.

Figure.R1. Dependence of the ratio of the apparent to simulated diffusion coefficients in a spherocylinder when analyzing the middle part and poles separately. Data from³ and⁴.

To further investigate the dependence of the apparent diffusion coefficient on the time separation, we measured the apparent diffusion coefficient of mEos3.2 in *E. coli* with Δt ranging from 1.5 to 9 ms. Fig. R2 shows the expected decrease of the measured diffusion coefficient with increasing Δt .

³ Buu Minh Tran et al., "Single-Protein Diffusion in the Periplasm of Escherichia Coli," *Journal of Molecular Biology* 436, no. 4 (February 2024): 168420, <https://doi.org/10.1016/j.jmb.2023.168420>.

⁴ Luca Mantovanelli et al., "Simulation-Based Reconstructed Diffusion Unveils the Effect of Aging on Protein Diffusion in Escherichia Coli," ed. Stefan Klumpp, *PLOS Computational Biology* 19, no. 9 (September 11, 2023): e1011093, <https://doi.org/10.1371/journal.pcbi.1011093>.

Figure R2. Dependence of the apparent diffusion coefficient of mEos3.2 in the middle part or pole regions of *E. coli* cells on the time separation (Δt) for the read-out laser pulses in a SMdM experiment.

Notably, even though the confinement effects result in an underestimation of the diffusion coefficients, it introduces the same systematic error irrespective of the experimental conditions. Hence, it does not influence the outcomes or conclusions from the experiments, which is also apparent from the ratio of the pole to middle diffusion coefficient as a function of Δt (Figure R3).

Figure R3. Dependence of the ratio between the apparent diffusion coefficient of mEos3.2 in the pole and middle regions of *E. coli* cells on Δt , i.e. the time separation of the read-out laser in the SMdM experiments.

6) L155 We propose that the decrease of DL at the E. coli poles and differences between old and new pole of the cell are caused by protein aggregates, which accumulate more at the old than new pole.

Figure S4 shows the representative probability density function fitting profiles of the displacements in the middle part, it would be important to have similar examples for the poles, at least one for each condition of Figure 3.

We added Supporting Figure S5 (see below) with representative probability density function fitting profiles of the displacements in the middle part and old and new poles of dividing and PopZ-expressing cells.

Fig. S5 Representative probability density function fitting profiles of the displacements in the middle part, old pole, and new pole of dividing *E. coli* BW25113-mEos3.2 and BW25113-popZ::eGFP, expressing PopZ-eGFP as a maker of the old pole. Equation (3), using maximum likelihood estimation, was used to fit the data. D_1 and b refer to lateral diffusion coefficient (D_1) and background correction coefficient, respectively.

For the main text, we added (lines 170 - 171):

“Representative probability density function fitting profiles of the displacements in different regions of the cells are shown in Figure S5.”

Following up on point (5), is SMDM, for a given cell, showing single peak probability density functions of the diffusion at the poles and in the middle (distribution with a single peak like in Figure 1F) or are we in presence of higher heterogeneity in the distributions at the poles? In this second case, would it be possible some of the proteins at the poles are immobile and some other moves at a similar rate than the ones in the cell middle? Could a FRAP measurement (more crude but still effective) assess the eventual presence of an immobile fraction or better help in excluding its

presence at the poles (this comment would be also important for the data in Figure 5)?

In a previous study, we introduced the analysis of two-component diffusion via SMdM. For example, in Fig. 6 of ⁵, the two-component analysis reveals an almost-immobile component in the diffusion of a blue-light receptor protein RsbL in *Listeria monocytogenes* cells.

For dividing and PopZ-expressing cells, we used the same two-component fitting procedure and always obtain a fraction of 1 for one of two components. We thus conclude that an immobile protein fraction is not present at the cell poles.

7) Figure 5. Could the authors add also representative fluorescence images under the brightfield ones?

We added PALM reconstruction images to panel C of Figure 5 (see new figure below) and “Super-resolution image reconstruction” section of the Methods section (lines 458-462).

⁵ Buu Minh Tran et al., “Super-Resolving Microscopy Reveals the Localizations and Movement Dynamics of Stressosome Proteins in *Listeria Monocytogenes*,” *Communications Biology* 6, no. 1 (January 14, 2023): 51, <https://doi.org/10.1038/s42003-023-04423-y>.

Figure 5. Induced protein aggregation slows down protein diffusion in *E. coli*. (A) Brightfield images of *E. coli* BW25113-*mEos3.2* with heat-induced protein aggregation. (B) Pole/middle D_L ratios for non-shocked and heat-shocked cells with one and two optically-dense aggregates. Data presented as mean \pm standard error of the mean (SEM), the number of non-shocked and shocked cells with one- and two-pole aggregates was 33, 23 and 20, respectively. (C) Widefield image, PALM reconstruction, displacement and diffusion maps for *E. coli* BW25113-*mEos3.2* with two (left) and one (right) pole with heat-induced aggregates. The pixel bin size of the displacement and diffusion maps is 50 nm. Color map for displacements represents the number of displacements per pixel. Diffusion maps were reconstructed by fitting displacements in each pixel bin with equation 3.

8) Figure 6: Overall, we find that areas with visual protein aggregates, triggered by the heat shock, had diminished data density and slower diffusion than areas without these aggregates (Fig. 6C and Fig. S8). The decrease in the diffusion coefficient is comparable to that in the pole regions of heat-shocked cells without cephalaxin treatment.

Could a fluorescence image be included in this figure and is it possible that acquiring with a different time step would change such percentages (Referring to “diminished data density”)?

We added the PALM reconstructed images to the supporting Figure S10. These reconstructions do not show a dependence of data density and diffusion on the length of the time-step used for the displacement measurements (see reply to question 5). These PALM reconstructions show lower data densities in regions where the division is blocked by the cephalaxin treatment.

Fig. S10. Displacement, diffusion maps and PALM reconstructions of *E. coli* BW25113-mEos3.2 cells treated for 7h with cephalaxin followed by a 1h heat-shock as described in the legend of Figure S9. The pixel bin size of the displacement and diffusion maps is 100 nm. Color map for displacement maps represents the number of displacements per pixel. Diffusion map reconstructed by fitting displacements in each pixel bin with equation (3). The numbers refer to the brightfield images shown in Figure S9.

The diminished displacement and localization density in cephalaxin-treated cells is described in the main text (lines 224 - 227) as follows:

“Overall, we find that areas with visual protein aggregates, triggered by the heat shock, had diminished displacement and localization density and slower diffusion than areas without these aggregates (Fig. 6C and Fig. S10).”

9) L229:

What is the actual spatial resolution in this case? Could the authors provide some kind of estimate? FCS (which, unlike FRAP, also belongs to the single molecule techniques family, this should be changed in the text) spatial resolution, by the way, is dictated by the confocal volume so it could also discriminate poles from middle.

For SMdM the spatial resolution of the reconstructed diffusion maps is determined by the bin size of the map, which can be selected arbitrarily and is 50 nm in our case.

The lower limit for the bin-size is determined by three main limitations:

- 1) To avoid averaging of the diffusion coefficients, the bin size should not be not much smaller than the displacements that are measured, which in our case are in the range of 100-200 nm. Thus, with a 50 nm bin-size we slightly oversample the data.
- 2) The number of displacements within the selected bins should be at least 10 for reliable fitting of bin-wise displacement distributions. In our data, hundreds of displacements are accumulated per 50 nm bin.
- 3) The measured displacements and the bin size should not be lower than the localization precision of single-molecule detection. In case of mEos3.2, the localization error was estimated to be ca. 12 nm (“Rational design of true monomeric and bright photoactivatable fluorescent proteins”, Nature Methods, 2012).

FCS can be referred to as a single-molecule technique; we modified the text (lines 95 - 98):

“In contrast to for instance, fluorescence correlation spectroscopy (FCS) and fluorescence recovery after photobleaching (FRAP), SMdM allows simultaneous acquisition of data in various regions of a cell or several adjacent cells, which allows the detection of a large number of molecules with a high precision and high throughput. “

10) L263 “However SMdM requires data accumulation for approximately 30 to 40 min during which the cell should be immobile”.

As this number is one-two order of magnitudes higher than the typical acquisition times for FCS or even FRAP measurements, could the author reassure me and the potential readers that both the middle part and the poles were immobile within this time range?

While the agarose pads would probably keep cells rather stable, I am worried that for very elongated cephalixin-treated cells one could see some wobbling of the poles in such a long time range. Could the authors at least include a couple of time-lapse movies in the supplementary material to show cells (especially elongated cells) are indeed immobile both on the xy plane and in the focal direction in such time range? Also, I feel that the information that one single measurement requires 30-40 minutes should be provided in the material and methods in case colleagues would want to

perform similar experiments, as this could be a bottleneck for some applications.

We recorded 40-min movies with a brightfield microscopy for untreated and cephalalexin-treated cells; cells were indeed immobile. The first and the last frames of the movie are shown below.

Figure R5. Brightfield images of the cell at the beginning and the end of a 40-min measurement. Top panel: untreated *E. coli* BW25113 at time 0 and after 40 min of immobilization on agarose pads. Bottom panel: cephalalexin-treated *E. coli* BW25113 at time 0 and after 40 min of immobilization on agarose pads.

To achieve a high stability of the sample, we turn on the microscope 4 hours before the actual measurements. We also apply continuous z-drift correction during data acquisition and apply xy-drift correction in the analysis.

We added the information about the z-drift correction in the “Data acquisition” section of the Methods (lines 412 - 415):

“The total measurement time for one field of view is approximately 30 to 40 mins. The autofocus function of the microscope was enabled to avoid z-drift. Within this time range, the cells were relatively immobile.”

11) L436 As a selection of pole regions of bacterial cells, we used 20 % of total

length of the (rotated) cells, and the remaining 60% as middle part where (most of) the nucleoid localizes. DAPI staining of *E. coli* cells showed that the nucleoid area is approximately 60 % of the cell length, that is, in cells growing exponentially on MBM supplemented with 0.1% glycerol (Fig. S10).

Bakshi et al. in 2014 showed a significantly smaller staining of the nucleoid using Sytox Orange, perhaps it would be good to rather base the distinction between nucleoid and poles on such dye localization, possibly acquiring also images with Sytox Orange staining?

We thank the Reviewer for the suggestion. We were not aware of the difference between Sytox Orange and DAPI. To quantify the difference in the diffusion between the middle and the pole of the cells, we used a volume for each pole of 20%. This estimate is based on the geometry of the average cell in the dataset, i.e. the radius of the cell is approximately 20% of the cell length (a cell radius of 0.45 μm is approximately 20% of the cell length of 2.25 μm). As an additional check, we now show by DAPI staining that the middle region of the cell largely overlaps with the nucleoid.

We edited the “Data analysis” section of Methods (lines 447 - 450) as follows:

“Based on the average dimensions of the cells, we used 20% of the total length of the (rotated) cells as pole region, and the remaining 60% as the middle part. DAPI staining of *E. coli* cells showed that the nucleoid-occupied area largely overlaps with the 60% middle part (Fig. S12)”

Minor Observation:

In the supplementary material, the authors write “Brightfield” images, in the main text they write sometimes “Widefield” (but “Brightfield directly in the figures), and differentiate fluorescence images calling them “Widefield Fluorescence” or just “Fluorescence”, I would advise for higher consistency to av

We made it consistent and kept “Brightfield” for images with transmitted light and “Widefield fluorescence” for fluorescence images throughout the text.

REVIEWERS' COMMENTS:

Reviewer #1 (Remarks to the Author):

The authors have made a good response to my comments on the previous version. I have no further modifications to suggest.

Reviewer #3 (Remarks to the Author):

The points raised were addressed satisfactorily.